# A High-Order Load Model and the Control Algorithm for an Aerospace Electro-Hydraulic Actuator [†]

**Shoujun Zhao** [1,2,*], **Keqin Chen** [1,2], **Xiaosha Zhang** [1,2], **Yingxin Zhao** [1,2], **Guanghui Jing** [1,2], **Chuanwei Yin** [1,2] **and Xue Xiao** [1,2]

1 Laboratory of Aerospace Servo Actuation and Transmission, China Aerospace Science and Technology Corporation, No.1 South DaHongMen Road, Fengtai District, Beijing 100076, China; ckq0315@163.com (K.C.); zxs602wwyx@163.com (X.Z.); zyx_0@sina.com (Y.Z.); jingghhit@163.com (G.J.); yinchw_calt@163.com (C.Y.); xiaoxue_ht@163.com (X.X.)

2 Beijing Institute of Precision Mechatronics and Controls, No.1 South DaHongMen Road, Fengtai District, Beijing 100076, China

* Correspondence: shoujunzhao@vip.sina.com; Tel.: +86-10-8852-0124

† This paper is an extended version of the conference paper by Shoujun Zhao, Keqin Chen, Xiaosha Zhang, Yingxin Zhao, Guanghui Jing, Chuanwei Yin, Xue Xiao. A Generalized Control Model and Its Digital Algorithm for Aerospace Electrohydraulic Actuators. In Proceedings of the 1st International Electronic Conference on Actuator Technology: Materials, Devices and Applications, 23–27 November 2020.

**Abstract:** It is difficult to describe precisely, and thus control satisfactorily, the dynamics of an electro-hydraulic actuator to drive a high thrust liquid launcher engine, whose structural resonant frequency is usually low due to its heavy inertia and complicated mass distribution, let alone one to drive a heavy kerolox engine with high-order dynamics. By transforming classic control block diagrams, a baseline two-mass-two-spring load model and a normalized actuator-engine system model were developed for understanding the basic physics and methodology, where a fourth-order transfer function is used to model the multi-resonance-frequency engine body outside of the rod position loop, another fourth-order transfer function with two pairs of conjugated zeros and poles to represent the composite hydro-mechanical resonance effect in the closed rod position loop. A sixth-order model was thereafter proposed for even higher dynamics. The model parameters were identified and optimized by a full factor search approach. To meet the stringent specification of static and dynamic performances, it was demonstrated that a notch filter network combined with other controllers is needed since the traditional dynamic pressure feedback (DPF) is difficult to handle the high-order dynamics. The approach has been validated by simulation, experiments and successful flights. The models, analysis, data and insights were elaborated.

**Keywords:** control model; control algorithm; high-order; electro-hydraulic actuator

## 1. Introduction

The electro-hydraulic servo actuation is a well-developed technology. Nonetheless, most of its physical understanding and mathematical modeling, from as early as 1967, have been referenced from the *Hydraulic Control Systems* by H.E. Merritt [1]. In most cases, it was assumed that the driven load had sufficiently high structural resonance frequencies which could be neglected, with only the hydraulic natural frequency remaining in the control loop. However, in aerospace applications, such as rudders, fins and engines, due to weight and space limitations, the structural stiffness is usually low but the required dynamics is high, and moreover, sometimes even high-order dynamics exhibits in the actuator loop. In such conditions, the model has to be modified and the load structural resonance should be included. This was discussed in the classic book and a general model with cascaded second-order transfer functions was used to depict loads with many degree-of-freedoms (DOF), where, however, only the dynamics at the motor shaft point

was elaborated with that at the load end left open for more work, and it was pointed out that because mechanical structures are continuous systems, they are described in partial differential equations of formidable complexity, and it is not presently possible to predict all the quantities involved [1] (pp. 157–162).

J.W. Edward presented an appropriate one-mass-one-spring model for an aircraft rudder control servo system [2]. A spring was inserted as the structural compliance into the interface between the piston rod and the rudder. The structural natural frequency, the hydraulic natural frequency and the derived composite hydro-mechanical natural frequency were clearly depicted and incorporated into the model, based on which the rudder surface dynamics was controlled by the dynamic pressure feedback (DPF) acting as a first-order high-pass filter and matched well with the test results. K.E. Rydberg promoted the concise approach in his *Hydraulic Servo Systems* [3] (pp. 35–37). However, in most current publications on electro-hydraulics, only the hydraulic resonance is included [4,5]. Furthermore, in most papers studying various emerging electro-hydrostatic actuators, where there are also hydraulic cylinders, only the hydraulic natural frequency is considered too [6,7]. This approach might work well elsewhere but not in highly dynamic aerospace actuators, especially for those to drive loads with high-order dynamics. J. Yoo demonstrated there are rich structural dynamics in a launcher engine [8]. Usually there is only one load resonance peak shown in the actuation loop and using one-mass-one-spring load model is sufficient [9–11]. For those showing higher order dynamics with heavy impact on actuators, there has been little literature.

Zhang C. showed that DPF remains welcome to suppress resonances in electro-hydraulic systems [12]. Nevertheless, with digital signal processors indispensable nowadays, a notch filter is preferred and proved effective [13–16], especially attractive in aerospace uses where hardware reduction means higher reliability. Since the fixedness of structural resonance frequencies can be guaranteed in aerospace engines, the application of a digital notch filter network also brings great conveniences in designs. On the other hand, the capacity of a traditional DPF is limited in dealing with the multiple load resonance peaks since only one peak can be treated with a first-order DPF filter. Therefore, multiple notch filters to regulate actuator dynamics need to be explored. Though Jang J. showed that such a method is adopted in flight control systems [17], it is an emerging topic in aerospace electro-hydraulic actuation systems.

Besides resonance suppression, superior static and tracking performances are required in aerospace actuators. This point is highlighted when a high power electro-hydraulic system shows heavier nonlinearity and its dynamics is not specified by a simple 3dB or $-45°$ bandwidth but by a set of amplitude and phase values at a series of selected frequencies. Usually, proportional-integral-differential (PID) controllers, or feedforward compensation, or the combination of both are used to deal with the problem [18–21]. Nonetheless, a more complex combination to include notch filters and these controllers is challenging for actuation designers.

The author's team has been working on actuators to gimbal non-toxic-non-pollution launcher engines [22–27]. To deal with the thrust vector control (TVC) of an engine which showed high-order structural dynamics, Yin C. introduced a two-mass-two-spring model and found that a fourth-order notch filter network control algorithm brought more effective results than DPF, nevertheless the process of developing the model was not explored and it was found hard to satisfy all the amplitude and phase requirements at the selected frequencies [26]. Zhao S. briefed a combined controller with satisfactory results in the conference paper but without details [27]. Deriving the model and dissecting the algorithm will be presented in this article.

The novel contributions of this paper are highlighted as follows: a baseline fourth-order or two-mass-two-spring load model for an electro-hydraulic actuator was elaborated, an even higher sixth-order model was thereafter proposed for the first time, its parameters were identified by a simple full factor search method, a combined control algorithm comprising cascaded notch filters, a feedforward compensation, a nonlinear PID controller

was fabricated, and the underlying regulating mechanism directed to responses in high, low and intermediate frequency bands was analyzed. It showcases the usefulness of a high-order model and a sophisticated controller for actuation systems in delicate applications.

The remaining paper is organized as follows. Section 2 introduces an aerospace electro-hydraulic actuation system and its modeling, including both a fourth-order and sixth-order load model. Section 3 describes the combined control algorithm. Section 4 briefs the model identification and parameter optimization by a full factor search method. Section 5 presents the experiment results. Finally, discussion and conclusions are given in Sections 6 and 7, respectively.

## 2. The System and Its Modeling

The assembly of an electro-hydraulic actuator and the driven 1200 kN kerolox engine is shown in Figure 1 [25,26]. The challenge is that the heavy turbo-pump assembly acts as a non-negligible minor mass connected to the main nozzle body so that two dominant resonance peaks emerge in the TVC loop and the traditional one-mass-one-spring model for the engine as the load of the actuator no longer applies. Additionally, as can be seen, there are some other smaller bodies outside of the main body, implying more minor resonance peaks in high frequency band.

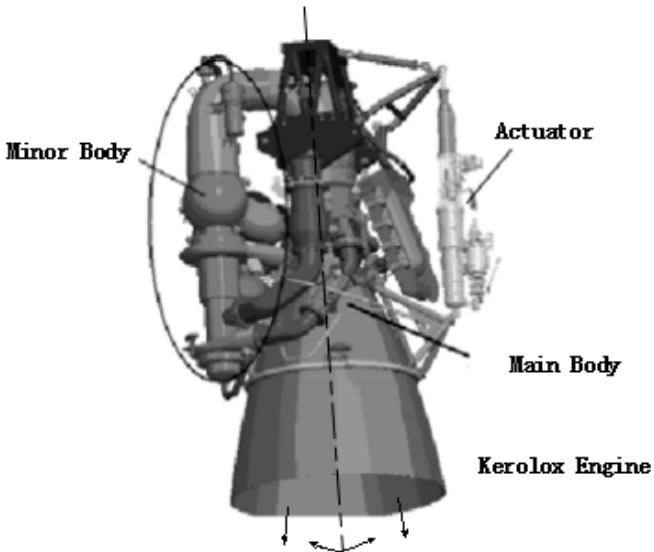

**Figure 1.** The assembly of the actuator and the driven kerolox engine.

A simplified schematics of the electro-hydraulic actuation system is described in Figure 2, where $Q_L$ is the load flowrate from the servo-valve to the actuator, $P_L$ is the differential pressure across the piston, $A_p$ is the acting piston area, $X_c$ is the normalized command signal, $X_p$ is the piston position, and $X_{L1}$ is the normalized load position output of the main body in the equivalent linear form. Usually, the engine gimbaling angle is not measured in flight but in ground tests in the form of linear displacement which is converted to the angular value.

The control loop includes a digital controller, a servo-valve, a double acting piston actuator and a linear displacement sensor embedded inside the rod. The controller closes the negative feedback loop and performs digital algorithms for static and dynamic compensations. The expected output is the engine's gimbaling thrust vector angle, i.e., $X_{L1}$. It is a classic aerospace actuator design, where the feedback signal is picked up via the sensor inside the piston rod rather than via the angular output sensor [2,28].

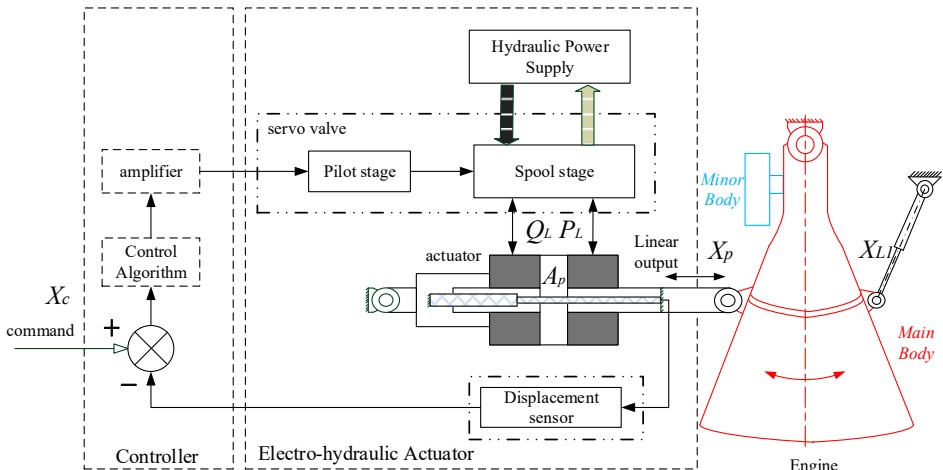

**Figure 2.** The simplified schematics for an aerospace electro-hydraulic actuation system.

For the high-order engine body's structural dynamics, a baseline two-mass-two-spring model was presented in Figure 3, where $M_1$, $K_{L1}$, $B_{L1}$, $X_{L1}$ and $M_2$, $K_{L2}$, $B_{L2}$, $X_{L2}$ are the equivalent mass, spring stiffness, viscous coefficient, linear displacement of the main engine body and the minor body, respectively, and the connected hydraulic actuator is given together.

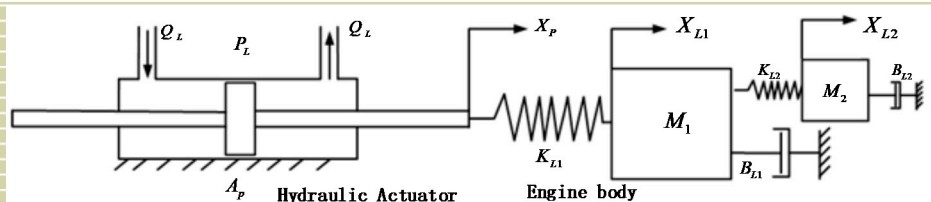

**Figure 3.** The physical model of the hydraulic actuator and the driven kerolox engine.

The mathematical equations are presented in Laplace-form Equations (1)–(7).

$$Q_v = \frac{K_{qi}}{\frac{s}{\omega_v} + 1} I_v \tag{1}$$

$$Q_L = Q_v - K_c P_L \tag{2}$$

$$Q_L = A_p s X_p + \frac{V_t}{4\beta} s P_L \tag{3}$$

$$A_p P_L = K_{L1}(X_p - X_{L1}) \tag{4}$$

$$K_{L1}(X_p - X_{L1}) = M_1 s^2 X_{L1} + B_{L1} s X_{L1} + K_{L2}(X_{L1} - X_{L2}) \tag{5}$$

$$K_{L2}(X_{L1} - X_{L2}) = M_2 s^2 X_{L2} + B_{L2} s X_{L2} \tag{6}$$

$$M = M_1 + M_2, \qquad M = \frac{J}{R^2} \tag{7}$$

where $s$ is the Laplace operator, $Q_v$ is the ideal servo-valve flowrate output, $K_{qi}$ is the nominal servo-valve flowrate gain, $I_v$ is the electrical current applied, $\omega_v$ is the first-order servo-valve frequency bandwidth, $K_c$ is the lumped leakage coefficient across the piston, including the internal leakage of the servo-valve, $V_t$ is the total control volume of the two actuator chambers, $\beta$ is the equivalent bulk modulus of the contained oil, $J$ is the lumped rotational load inertia, and $R$ is the nominal rotation radius of the load.

The servo-valve is modeled as a first-order transfer function as in Equation (1), whose bandwidth is chosen to be much higher than the structural resonance, and where nonlin-

earity is neglected first for a clearer presentation but accommodated later by the control algorithm, such as curved gains at the null spool region and the friction forces. Equation (2) gives the load flow from the servo-valve to the cylinder by subtracting the lumped leakages. Equation (3) presents the load flow in the form of consumption by the cylinder, including the effective flowrate to move the piston and the compressed terms by the hydraulic pressure. Equation (4) is that of the force balance at the piston node, where the moving piston mass is neglected due to its much smaller magnitude than the engine. Equation (5) is that of the force balance at the main mass body node, where the inertia force, equivalent spring forces and viscous forces are included, but with other minor items neglected, such as the bearing friction, the bellows elastic load and the biased thrust torque, etc. Equation (6) is that of the force balance at the minor mass body node. Equation (7) converts rotational inertia to linear mass.

It is noteworthy that, in the real systems, there are not real connection springs, even though real springs exist in load simulators. Additionally, in practice, the system is designed in such a way that other structures, like the piston rod, the actuator body and the fixed supporting structures, are so strong that only the stiffness inside the load needs to be modeled.

To derive a final normalized model, the control block diagram reduction approach is the most straightforward and clearest.

The first step is that Equations (5) and (6) are transformed into transition functions as Equations (8) and (9), with structural natural frequencies and damping ratios given in Equations (10) and (11), and the equivalent structural stiffness ratio in Equation (12).

$$X_p = \left( \frac{s^2}{\omega_{L1}^2} + \frac{2\xi_{L1}}{\omega_{L1}}s + 1 \right) X_{L1} + \alpha(X_{L1} - X_{L2}) \tag{8}$$

$$X_{L1} = \left( \frac{s^2}{\omega_{L2}^2} + \frac{2\xi_{L2}}{\omega_{L2}}s + 1 \right) X_{L2} \tag{9}$$

$$\omega_{L1} = \sqrt{\frac{K_{L1}}{M_1}}, \quad \omega_{L2} = \sqrt{\frac{K_{L2}}{M_2}} \tag{10}$$

$$\xi_{L1} = \frac{B_{L1}}{2K_{L1}}\sqrt{\frac{K_{L1}}{M_1}}, \ \xi_{L2} = \frac{B_{L2}}{2K_{L2}}\sqrt{\frac{K_{L2}}{M_2}} \tag{11}$$

$$\alpha = \frac{K_{L2}}{K_{L1}} \tag{12}$$

where ($\omega_{L1}$, $\xi_{L1}$, $\omega_{L2}$, $\xi_{L2}$) are structural resonant frequencies and the corresponding damping ratios arising for the main body and the minor body, respectively, and $\alpha$ is the structural stiffness ratio.

Note that the engine's structural resonant frequencies ($\omega_{L1}$, $\omega_{L2}$) and the damping ratios ($\xi_{L1}$, $\xi_{L2}$) are independent of any other electro-hydraulic actuator design, except for the installation geometry on which the rotation radius R depends. In ground testing, there are two measurement points, one at the piston rod as $X_p$ and the other at the engine gimbaling angular output as $X_{L1}$.

The next step is to present Equations (8) and (9) into a control block diagram, as in Figure 4.

Since only $X_{L1}$ is expected, with the minor body angular output $X_{L2}$ eliminated, the transfer function blocks from $X_p$ to $X_{L1}$ are derived into Figure 5, i.e., the engine thrust vector output dynamics. With the feedback loop furtherly eliminated, the condensed form is given in Figure 6.

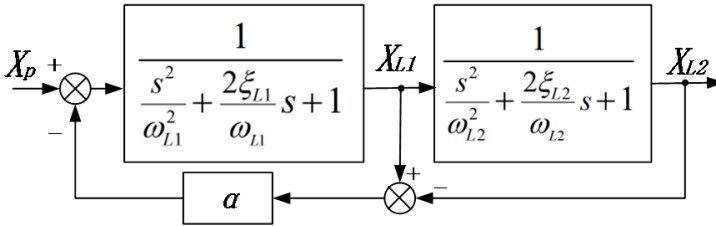

**Figure 4.** The control block diagram presentation of Equations (8) and (9).

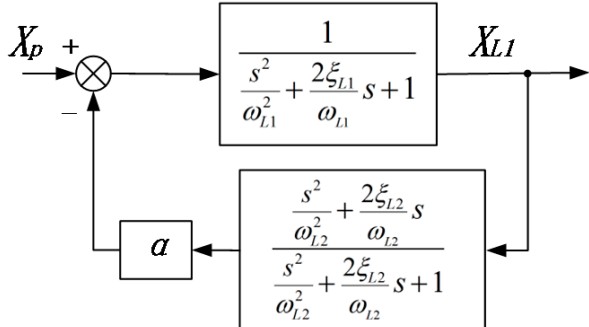

**Figure 5.** The control block diagram of the thrust vector dynamics.

$$X_p \quad \cfrac{\dfrac{s^2}{\omega_{L2}^2} + \dfrac{2\xi_{L2}}{\omega_{L2}}s + 1}{\left(\dfrac{s^2}{\omega_{L1}^2} + \dfrac{2\xi_{L1}}{\omega_{L1}}s + 1\right)\left(\dfrac{s^2}{\omega_{L2}^2} + \dfrac{2\xi_{L2}}{\omega_{L2}}s + 1\right) + \alpha\left(\dfrac{s^2}{\omega_{L2}'^2} + \dfrac{2\xi_{L2}}{\omega_{L2}}s\right)} \quad X_{L1}$$

**Figure 6.** The condensed control block diagram of the thrust vector dynamics.

For a more condensed representation, the denominator in Figure 6 has to be written into a standard form of cascaded second-order transfer functions, let:

$$\left(\frac{s^2}{\omega_{L1}'^2} + \frac{2\xi_{L1}'}{\omega_{L1}'}s + 1\right)\left(\frac{s^2}{\omega_{L2}'^2} + \frac{2\xi_{L2}'}{\omega_{L2}'}s + 1\right)$$
$$= \left(\frac{s^2}{\omega_{L1}^2} + \frac{2\xi_{L1}}{\omega_{L1}}s + 1\right)\left(\frac{s^2}{\omega_{L2}^2} + \frac{2\xi_{L2}}{\omega_{L2}}s + 1\right) + \alpha\left(\frac{s^2}{\omega_{L2}^2} + \frac{2\xi_{L2}}{\omega_{L2}}s\right) \tag{13}$$

In Equation (13), a normalized transfer function from the piston position to the thrust vector angular output is depicted and furtherly represented in the form of block diagram as in the Figure 7, where the derived equivalent resonance frequencies and damping ratios $(\omega_{L1}', \xi_{L1}', \omega_{L2}', \xi_{L2}')$ replaced the original ones $(\omega_{L1}, \omega_{L2}, \xi_{L1}, \xi_{L2})$.

$$X_p \quad \cfrac{\dfrac{s^2}{\omega_{L2}^2} + \dfrac{2\xi_{L2}}{\omega_{L2}}s + 1}{\left(\dfrac{s^2}{\omega_{L1}'^2} + \dfrac{2\xi_{L1}'}{\omega_{L1}'}s + 1\right)\left(\dfrac{s^2}{\omega_{L2}'^2} + \dfrac{2\xi_{L2}'}{\omega_{L2}'}s + 1\right)} \quad X_{L1}$$

**Figure 7.** The normalized control block diagram of the thrust vector dynamics.

To obtain the system dynamics that comprise the actuator and the engine, Equations (2)–(4) are included to give Figure 8.

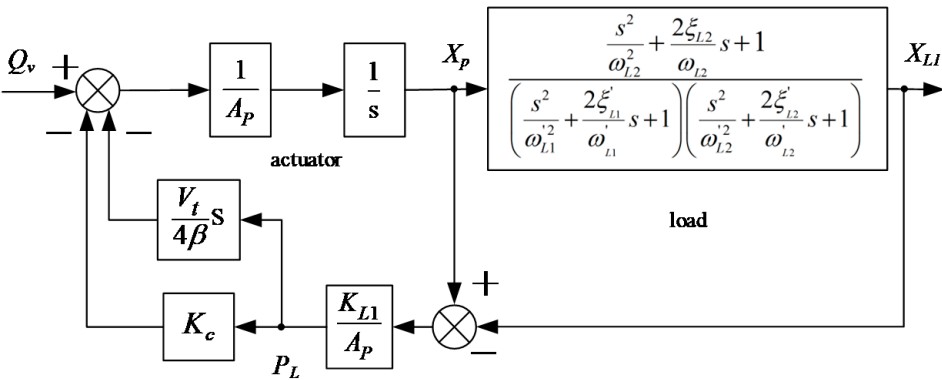

**Figure 8.** The control block diagram of actuator-engine system.

With minor loops eliminated and some algebraic transformations, the condensed actuator-engine diagram is represented in Figure 9, with the two measurement points $X_p$ and $X_{L1}$ remained.

**Figure 9.** The condensed control block diagram of the actuator-engine system.

In Figure 9, the composite hydro-mechanical resonance frequencies ($\omega_{c1}$, $\omega_{c2}$) are coupled from the structural resonant frequencies ($\omega'_{L1}$, $\omega'_{L2}$) and hydraulic resonant frequencies ($\omega_{h1}$, $\omega_{h2}$). Analogous to the one-mass-one-spring model [2,3,27], the first set of hydraulic and composite frequencies ($\omega_{h1}$, $\omega_{c1}$) can be given in Equation (14), similar forms of the second set of frequencies ($\omega_{h2}$, $\omega_{c2}$) are proposed in Equations (15) and (16), where $k_{\omega h2}$ is a correction factor, and the damping ratios ($\xi_{c1}$, $\xi_{c2}$) have to be presented as ($f_{\xi c1}$, $f_{\xi c2}$) roughly, since a direct derivation is impossible. In fact, their precise algebraic representations are not cared much since they can be identified from experiment as shown later.

$$\omega_{h1} = \sqrt{\frac{4A_p^2\beta}{M_1 V_t}} , \quad \frac{1}{\omega_{c1}^2} = \frac{1}{\omega'^2_{L1}} + \frac{1}{\omega_{h1}^2} \tag{14}$$

$$\omega_{h2} = k_{\omega h2}\sqrt{\frac{4A_p^2\beta}{M_2 V_t}} \tag{15}$$

$$\frac{1}{\omega_{c2}^2} = \frac{1}{\omega'^2_{L2}} + \frac{1}{\omega_{h2}^2} \tag{16}$$

$$\xi_{c1} = f_{\xi c1}\left(\omega'_{L1}, \omega'_{L2}, \xi'_{L1}, \xi'_{L2}, V_t, K_{L1}, K_c, A_p, \beta\right) \tag{17}$$

$$\xi_{c2} = f_{\xi c2}\left(\omega'_{L1}, \omega'_{L2}, \xi'_{L1}, \xi'_{L2}, V_t, K_{L1}, K_c, A_p, \beta\right) \tag{18}$$

With the servo-valve dynamics in Equation (1) and a controller added, as well as a lumped open loop gain $K_o$, the final normalized system model is given in Figure 10, where $e(t)$ is the normalized position error between $X_c$ and $X_p$.

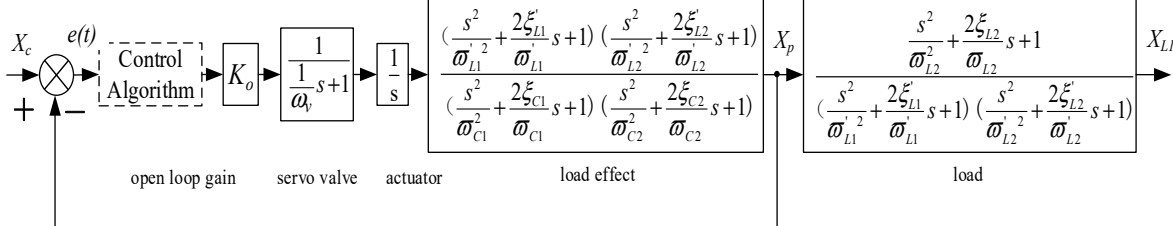

**Figure 10.** A normalized block diagram of the aerospace electro-hydraulic actuation system.

The open-loop $K_o$ is represented in Equation (19).

$$K_o = \frac{K_p K_{vi} K_{qi} K_{xf}}{A_p} \tag{19}$$

where $K_p$ is the nominal error amplification gain, $K_{vi}$ is the lumped voltage-to-ampere conversion coefficient of the digital-to-analog (D/A) converter and the servo-valve coil driver, $K_{xf}$ is the lumped conversion coefficient of the analog-to-digital (A/D) converter and the feedback displacement sensor.

As can be seen in Figure 10, for the expected output $X_{L1}$, it is only a half-closed loop. It needs to note that the half-closed loop is a classic aerospace design [2,28], where the actuator acts as an integrated control device for the simplest design and therefore the most reliable reasons in a higher system perspective, rather than another angular sensor needed in flight, though D.V. Lazić studied the approach [4]. Inside the piston position $X_p$ loop, the fourth-order transfer function with two pairs of zeros and poles is dominant, both poorly damped, representing the effect of the outside fourth-order load dynamics on the closed position loop, called "load effect", which is more complicated than that of an ordinary one-mass-one-spring modeled system [2,3,26].

Additionally, as can be imagined, there should be higher order models for some actuation systems, like sixth or eighth order, which are conceivably much harder to derive directly. As an analogy to the above fourth-order, a system block diagram incorporating a sixth-order or three-mass-three-spring load model was proposed in Figure 11, where $(\omega'_{L3}, \xi'_{L3})$ are the third derived load structural resonance frequency and its damping ratio, $(\omega_{c3}, \xi_{c3})$ are the corresponding third hydro-mechanical resonance frequency and its damping ratio, and $(\omega''_{L2}, \xi''_{L2}, \omega''_{L3}, \xi''_{L3})$ are the derived resonance frequencies and their damping ratios in the numerator, which are different from $(\omega_{L2}, \xi_{L2})$ as in a two-mass-two-spring model, since they can be not directly derived as in preceding Equation (10).

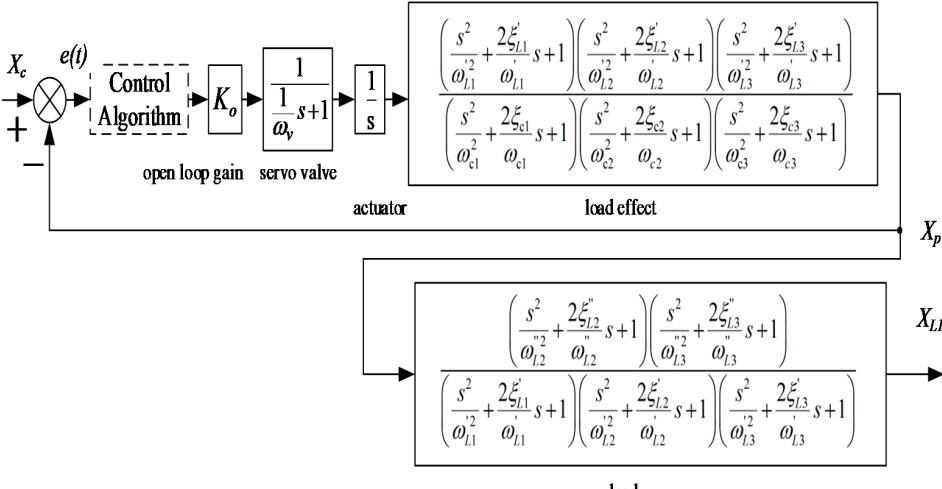

**Figure 11.** A normalized block diagram of the aerospace electro-hydraulic actuation system that incorporates a sixth-order load model.

For the high dynamics, a big acting area $A_p$ is needed to guarantee the composite hydro-mechanical frequencies are only slightly smaller than the structural frequencies, e.g., 10%. However, as for the heavy kerolox engine, since its structural natural frequencies are inevitably low, their effect on the system cannot be neglected as in common applications.

Moreover, in such a highly demanding system, the dynamics is not evaluated by a simple $-3$ dB or $-45°$ bandwidth but by a set of amplitude and phase values at a series of specified frequencies. Therefore, this clear understanding and precise depiction of the system dynamics are indispensable and a simple control approach to use only separate DPF, notch filter or PID controller cannot satisfy the stringent requirements.

### 3. The Combined Control Algorithm

A combined control strategy was developed as in Figure 12, where $G_n$, $K_p(e(t)) - K_i(e(t)) - K_d(e(t))$ and $F(X_c)$ represent the notch filter network, PID and feedforward controller, respectively.

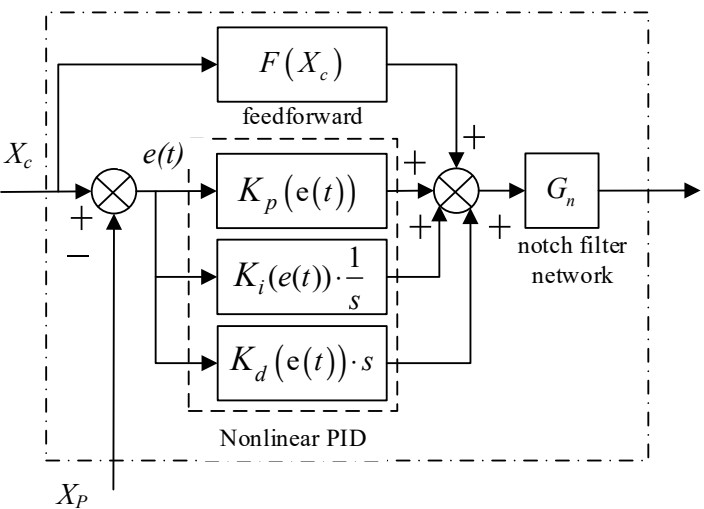

**Figure 12.** The combined control strategy.

The control algorithm comprises three parts: a notch filter network, a nonlinear PID and a feedforward compensation.

A baseline two-notch-filter network $G_n$ is shown as Equation (20).

$$G_n = \frac{\frac{1}{\omega_{n1}^2}s^2 + \frac{2\xi_{n1}}{\omega_{n1}}s + 1}{\frac{1}{\omega_{d1}^2}s^2 + \frac{2\xi_{d1}}{\omega_{d1}}s + 1} \cdot \frac{\frac{1}{\omega_{n2}^2}s^2 + \frac{2\xi_{n2}}{\omega_{n2}}s + 1}{\frac{1}{\omega_{d2}^2}s^2 + \frac{2\xi_{d2}}{\omega_{d2}}s + 1} \tag{20}$$

The rationale is that the poorly damped poles $(\omega_{c1}, \xi_{c1})$ and $(\omega_{c2}, \xi_{c2})$ are cancelled out by a pair of nearby poorly damped zeros $(\omega_{n1}, \xi_{n1})$ and $(\omega_{n2}, \xi_{n2})$ and replaced with a pair of better damped poles $(\omega_{d1}, \xi_{d1})$ and $(\omega_{d2}, \xi_{d2})$. In aerospace applications, the structural quality is guaranteed so that the resonance frequencies are controlled well within tolerances and notch filters can be applied in faith. On the other hand, the parameters can be optimized to change the width and depth of the notch window to accommodate permitted model variations, even to control a three resonance peak model as later shown. For non-stationary natural frequencies, Yao J. recommended adaptive notch filters [16].

Yin C. indicated that the capacity of a traditional DPF is limited in dealing with this kind of high-order load dynamics since only one resonance peak can be treated with the first-order DPF high-pass filter as in Figure 13 [26], where $K_{dpf}$ is the gain and $\tau$ is the time constant. Therefore, a multiple-notch-filter network is an indispensable choice rather than a replaceable one.

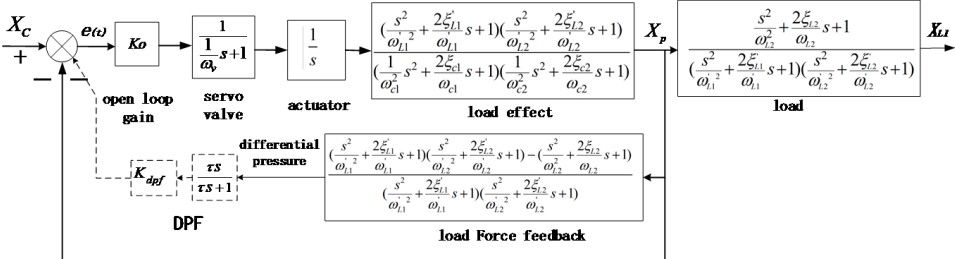

**Figure 13.** The dynamic pressure feedback (DPF) controlled system model illustration.

A nonlinear PID is used to improve the tracking accuracy in the low frequency band, i.e., around 1~5 rad/s, with a piecewise proportional gain presented as Equation (21).

$$K_P(e(t)) = \begin{cases} f_K \cdot K_P \ (f_K > 1) & |e(t)| \le e_n \\ K_P & |e(t)| > e_n \end{cases} \tag{21}$$

where $f_K$ is the bigger portion factor of the piecewise proportional gain and $e_n$ is prescribed error threshold under which the bigger gain is used.

A higher gain near zero is used to deal with the nonlinearity at the servo-valve spool center that is not modeled here. The differential factor is usually small and helps to overcome the un-modeled stiction in the system.

As to the integral factor, it helps to improve the positioning precision but needs to be designed elaborately as in Equations (22)–(25).

$$K_i = K_{i1} K_{i2} K_{i3} K_{in} \tag{22}$$

$$K_{i1} = \begin{cases} 0 & |P_s(t)| \le 0.9 P_{sn} \\ 1 & |P_s(t)| > 0.9 P_{sn} \end{cases} \tag{23}$$

$$K_{i2} = \begin{cases} 1 & |X_p(t)| \le 0.95 X_{pmax} \\ 0 & |X_p(t)| > 0.95 X_{pmax} \end{cases} \tag{24}$$

$$K_{i3} = \begin{cases} 1 & |e(t)| \le a \\ \frac{a+b-e(t)}{b} & a \le |e(t)| < b \\ 0 & |e(t)| > b \end{cases} \tag{25}$$

where $K_{in}$ is the nominal integral gain, $K_{i1}$ is the on-off switch triggered by the hydraulic power supply, $P_s(t)$ and $P_{sn}$ are the instant and nominal supply pressure, respectively, $K_{i2}$ is the on-off switch triggered by the maximum piston stroke, $X_{pmax}$ is the maximum piston stroke, $K_{i3}$ is the piecewise gain, and $a$ and $b$ are the constants to regulate the integral gain near and far from zero.

Equation (23) is used to set the integral active only when the hydraulic power is on so that an integration saturation and thus a jittering at the startup can be avoided. Similarly, Equation (24) is used to close the integral function at the maximum stroke so that a jittering at the position reversal near the rod ends can be prevented. Equation (25) is used to set the integral factor bigger near zero position error but smaller even zero in fast movements to exploit its maximum benefits and eliminate its side effects simultaneously.

An ideal feedforward compensation for the fourth-order system is the inverse plant model as given in Equation (26). However, due to the high-order differentiation, too much unwelcome noise would be introduced. Therefore, a reduced-order feedforward function is used as in Equation (27), where $K_f$ is the feedforward gain.

$$F(X_c) = \frac{\left(\frac{1}{\omega_v}s+1\right) \cdot \left(\frac{1}{\omega_{c1}^2}s^2 + \frac{2\xi_{c1}}{\omega_{c1}}s + 1\right)\left(\frac{1}{\omega_{c2}^2}s^2 + \frac{2\xi_{c2}}{\omega_{c2}}s + 1\right)}{K_o \cdot \left(\frac{s^2}{\omega_{L1}'^2} + \frac{2\xi_{L1}'}{\omega_{L1}'}s + 1\right)\left(\frac{s^2}{\omega_{L2}'^2} + \frac{2\xi_{L2}'}{\omega_{L2}'}s + 1\right)} \cdot s \cdot X_c \tag{26}$$

$$F = K_f \cdot s \cdot X_c \tag{27}$$

It is to point out that, also due to the existing higher order dynamics, to prevent high gains to emerge in high frequency band which would drive unwanted resonances, a large feedforward factor cannot be used. Therefore, a reasonably small gain $K_f$ helps to improve tracking precision in the intermediate frequency band.

As described, with the accurate depiction of the plant dynamics and the delicate use of a combined control strategy, the system vibration due to structural resonance should be well suppressed while a satisfactory frequency response in the whole frequency region from low to high be obtained.

## 4. Model Identification

For an electro-hydraulic servo actuator system to drive a high thrust kerolox launcher engine, the main parameters are shown in Table 1. Since the sampling time (digital control cycle) 0.001 second is so small compared to the system response that the negative digitization effect is neglected here.

**Table 1.** The main parameters of an actuation system.

| Parameter | Symbol | Value | Unit |
|---|---|---|---|
| Lumped engine rotational inertia | $J$ | 1304 | kg.m$^2$ |
| Engine rotational arm | $R$ | 845 | mm |
| Actuator acting piston area | $A_p$ | 4398 | mm$^2$ |
| System pressure | $P_s$ | 24 | MPa |
| Servo-valve bandwidth ($-45°$) | $\omega_v$ | $\geq 180$ | rad/s |
| Nominal open loop gain | $K_o$ | $\approx 20$ | rad/s |
| Maximum angular output | - | 8 | degree |
| Digital Control Cycle | - | 0.001 | second |

With an open loop gain as small as 15 rad/s and without any other compensation, given a series of sinusoidal commands, the system was tested to investigate the load resonance, with the amplitude bode plot of $X_{L1}$ and $X_p$ shown in Figure 14.

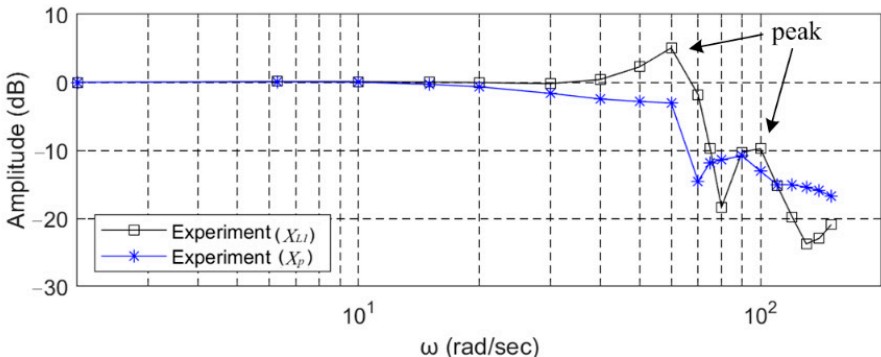

**Figure 14.** The dynamics of an uncompensated electro-hydraulic actuator and its engine load.

It is worth noting that, without compensations, with a relatively small open loop gain, the system vibrates seriously, as shown by the amplitude peaks of the load response $X_{L1}$ in Figure 14. It is clear that the system has to be compensated for a bigger gain and hence better dynamics.

The engine's structural resonance dynamics can be obtained by directly subtracting the response of $X_p$ at the piston point from that of $X_{L1}$ at the load output. The resulted amplitude curve is plotted in Figure 15, where both an approximate fourth-order and sixth-order model are plotted together.

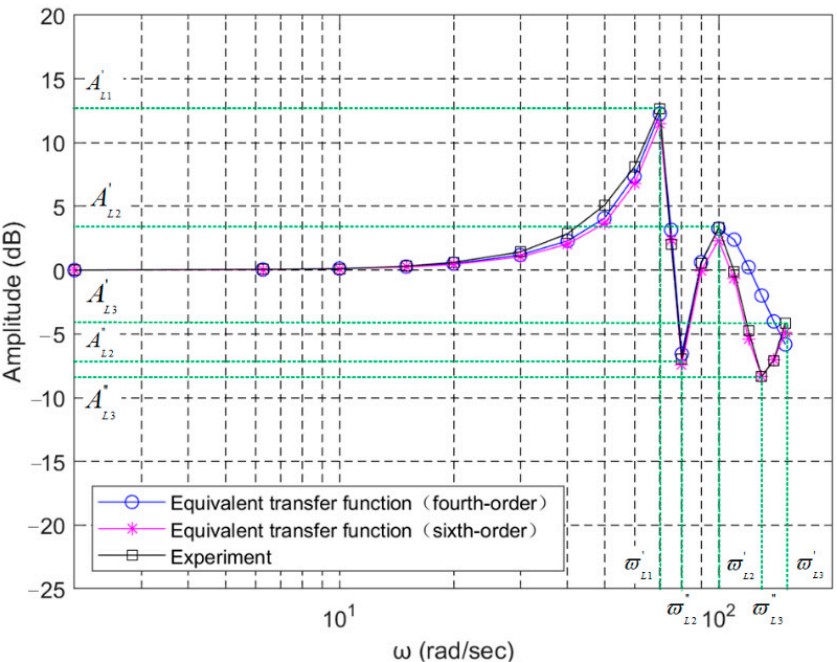

**Figure 15.** The structural dynamics of a kerolox launcher engine.

In the curves, there are two obvious denominator resonance frequencies as indicated by the two peaks $(A'_{L1}, A'_{L2})$ with $\omega'_{L1} = 70$ rad/s and $\omega'_{L2} = 100$ rad/s, and two obvious numerator resonance frequencies as indicated by the two bottoms $(A''_{L2}, A''_{L3})$ with $\omega''_{L2} = 80$ rad/s and $\omega''_{L3} = 130$ rad/s. Additionally, a third resonance frequency $\omega'_{L3}$ is implied by the upwardly bent curve above 130 rad/s, though its peak is not shown due to untested higher frequency points.

To identify the parameters, a simplest full factor search optimization method was used since it is the most straightforward and the computing time is acceptable, in spite of other advanced methods available. With the approximate fourth-order model as an example, the method is given as follows.

Firstly, the resonance frequencies can be easily found at the peaks and the bottoms, and the remaining three damping ratio parameters $\left(\xi'_{L1}, \xi'_{L2}, \xi''_{L2}\right)$ are given three sets of discrete values in an estimated scope around the guessed points as follows:

$\xi'_{L1} = (0.03 \sim 0.06)$, the interval is 0.001;
$\xi'_{L2} = (0.10 \sim 0.16)$, the interval is 0.001;
$\xi''_{L2} = (0.01 \sim 0.04)$, the interval is 0.001.

Each model choice is computed by programming to give the amplitude values at the selected frequencies.

Secondly, a set of amplitude difference tolerances are chosen, to judge the differences between the model and the experiment in the interested frequencies as in Table 2.

As long as its amplitude differences are located in the specified scope, a model is picked out, collected into the second but smaller set of models.

Thirdly, the smallest composite amplitude difference as in Equation (28) is chosen as the final criteria to select out the optimized model $(\xi'_{L1}, \xi'_{L2}, \xi''_{L2})$ from the second set.

$$\Delta A_{Lc} = 0.5\Delta A'_{L1} + 0.2\Delta A''_{L2} + 0.3\Delta A'_{L2} \tag{28}$$

where $\Delta A_{Lc}$ is the composite amplitude difference, $\Delta A'_{L1}$, $\Delta A''_{L2}$ and $\Delta A'_{L2}$ are the amplitude differences at peak and bottom frequencies, namely 70 rad/s, 80 rad/s and 100 rad/s, and 0.5, 0.2 and 0.3 are the weighting factors.

**Table 2.** The amplitude value tolerances at the selected frequencies.

| Frequency (rad/s) | Amplitude Value Tolerance (dB) |
|:---:|:---:|
| 60 | 1 |
| 70 | 2 |
| 75 | 2 |
| 80 | 3 |
| 90 | 3 |
| 100 | 2 |
| 110 | 3 |
| 130 | 5 |
| 140 | 8 |
| 150 | 8 |
| 160 | 8 |

Similarly, the model parameters for the sixth-order model can also be decided.

The identified parameters of the fourth-order and sixth-order model are listed in Tables 3 and 4, respectively, also with composite hydro-mechanical resonance frequencies and damping ratios given. It is to note that in the fourth-order model, $(\omega''_{L2}, \xi''_{L2})$ are used same as in the sixth-order rather than $(\omega_{L2}, \xi_{L2})$ in the original fourth-order as in the preceding Figure 7, because it is a reduced form from the sixth-order.

**Table 3.** The identified parameters of the fourth-order load model.

| Parameter | Symbol | Value | Unit |
|:---|:---:|:---:|:---:|
| Derived equivalent main structural resonance frequency | $\omega'_{L1}$ | 70 | rad/s |
| Derived equivalent main resonance damping ratio | $\xi'_{L1}$ | 0.05 | – |
| Derived equivalent minor structural resonance frequency | $\omega'_{L2}$ | 100 | rad/s |
| Derived equivalent minor resonance damping ratio | $\xi'_{L2}$ | 0.186 | – |
| Numerator resonance frequency | $\omega''_{L2}$ | 80 | rad/s |
| Numerator resonance damping ratio | $\xi''_{L2}$ | 0.034 | – |
| First composite hydro-mechanical resonance frequency | $\omega_{c1}$ | 67 | rad/s |
| First composite hydro-mechanical damping ratio | $\xi_{c1}$ | 0.055 | – |
| Second composite hydro-mechanical resonance frequency | $\omega_{c2}$ | 85 | rad/s |
| Second composite hydro-mechanical resonance damping ratio | $\xi_{c2}$ | 0.15 | – |

**Table 4.** The identified parameters of the sixth-order load model.

| Parameter | Symbol | Value | Unit |
|:---|:---:|:---:|:---:|
| First derived equivalent structural resonance frequency | $\omega'_{L1}$ | 70 | rad/s |
| First derived equivalent resonance damping ratio | $\xi'_{L1}$ | 0.05 | – |
| Second derived equivalent structural resonance frequency | $\omega'_{L2}$ | 100 | rad/s |
| Second derived equivalent resonance damping ratio | $\xi'_{L2}$ | 0.14 | – |
| Third derived equivalent structural resonance frequency | $\omega'_{L3}$ | 158 | rad/s |
| Third derived equivalent resonance damping ratio | $\xi'_{L3}$ | 0.19 | – |
| First numerator resonance Frequency | $\omega''_{L2}$ | 80 | rad/s |
| First numerator resonance damping ratio | $\xi''_{L2}$ | 0.032 | – |
| Second numerator resonance frequency | $\omega''_{L3}$ | 130 | rad/s |
| Second numerator resonance damping ratio | $\xi''_{L3}$ | 0.09 | – |
| First composite hydro-mechanical resonance frequency | $\omega_{c1}$ | 67 | rad/s |
| First composite hydro-mechanical damping ratio | $\xi_{c1}$ | 0.055 | – |
| Second composite hydro-mechanical resonance frequency | $\omega_{c2}$ | 85 | rad/s |
| Second composite hydro-mechanical resonance damping ratio | $\xi_{c2}$ | 0.15 | – |
| Third composite hydro-mechanical resonance frequency | $\omega_{c3}$ | 145 | rad/s |
| Third composite hydro-mechanical resonance damping ratio | $\xi_{c3}$ | 0.1 | – |

As shown in Figure 15, the load resonances can be modeled as a sixth-order transfer function with a sufficient precision, while a fourth-order function can match well the first and second peak but not the third peak.

To observe the hydro-mechanical resonance effect, the open loop response $X_{po}$ is computed by breaking the closed loop response $X_p$, as shown in Equations (29)–(32).

$$C_{xp} = 10^{\frac{A_{xp}}{20}}\left(\cos\theta_{xp} + i\cdot\sin\theta_{xp}\right) \tag{29}$$

$$C_{xpo} = \frac{C_{xp}}{1 - C_{xp}} \tag{30}$$

$$A_{xpo} = 20log_{10}|C_{xpo}| \tag{31}$$

$$\theta_{xpo} = angle(C_{xpo}) \tag{32}$$

where $C_{xp}$ and $C_{xpo}$ are the closed-loop and the open-loop frequency responses of the piston position $X_p$ represented in the complex form, and $A_{xp}$, $\theta_{xp}$, $A_{xpo}$, $\theta_{xpo}$ are corresponding amplitude and phase responses, respectively.

Equation (29) changes the closed-loop amplitude and phase values into complex numbers, Equation (30) computes the open-loop complex numbers by the closed-loop ones, and Equations (31) and (32) give the open-loop amplitude and phase responses.

The calculated open-loop amplitude bode plot is given in Figure 16.

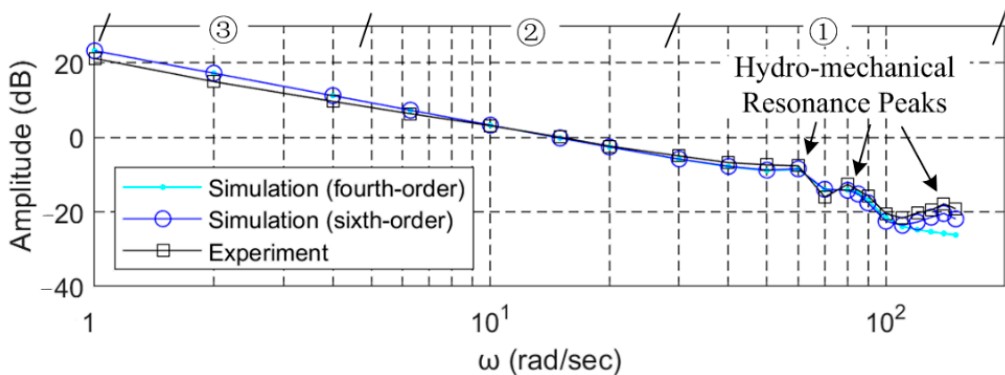

**Figure 16.** The open loop dynamics of the piston position $X_p$ loop.

In Figure 16, two obvious hydro-mechanical resonance peaks can be seen clearly and the third also emerges since its composite frequency $\omega_{c3}$ is 145 rad/s, lower than the corresponding load structural frequency $\omega'_{L3}$ and located in the plotted frequency scope. As shown, a sixth-order model fits the data better than a fourth-order.

Additionally, in Figure 16, the combined control strategy can be illustrated in the three marked regions. While the notch filters deal with the resonances in the high frequency region ①, the feedforward improves the tracking precision in the intermediate frequency region ②, and the PID upgrades the response in the low frequency region ③.

## 5. Experiments

The combined control algorithm as in Section 3 was used, with the parameters shown in Table 5, where the open loop gain was settled at 20 rad/s. The bode plot of the notch filter network is given in Figure 17. As shown, a two-notch-filter network was used, with the second notch frequency selected at 135 rad/s, in the middle of the second and third peak, i.e., 100 rad/s and 158 rad/s, with a width of around 100 rad/s, the lowest depth at −18 dB, and the shallowest depth at −5 dB. A three-notch-filter network was shown together, with a better amplitude performance, but with too much phage lag in the intermediate frequency region and more complexities, not used so far.

**Table 5.** The principal parameters of the combined control algorithm.

| Parameter | Symbol | Value | Unit |
| --- | --- | --- | --- |
| Nominal open-loop gain | $K_o$ | 20 | rad/s |
| Nominal proportional factor | $K_P$ | 1.032 | – |
| The enlargement proportional factor near zero error | $f_K$ | 1.5 | – |
| Nominal integral factor | $K_i$ | 0.1 | – |
| Nominal differential factor | $K_d$ | 0.006 | – |
| Feedforward factor | $K_f$ | 0.005 | – |
| Notch filter Resonance Frequency | $\omega_{n1}$ | 67 | rad/s |
| Notch filter Damping Ratio | $\xi_{n1}$ | 0.03 | – |
| Notch filter Resonance Frequency | $\omega_{d1}$ | 70 | rad/s |
| Notch filter Damping Ratio | $\xi_{d1}$ | 0.4 | – |
| Notch filter Resonance Frequency | $\omega_{n2}$ | 135 | rad/s |
| Notch filter Damping Ratio | $\xi_{n2}$ | 0.04 | – |
| Notch filter Resonance Frequency | $\omega_{d2}$ | 135 | rad/s |
| Notch filter Damping Ratio | $\xi_{d2}$ | 0.35 | – |

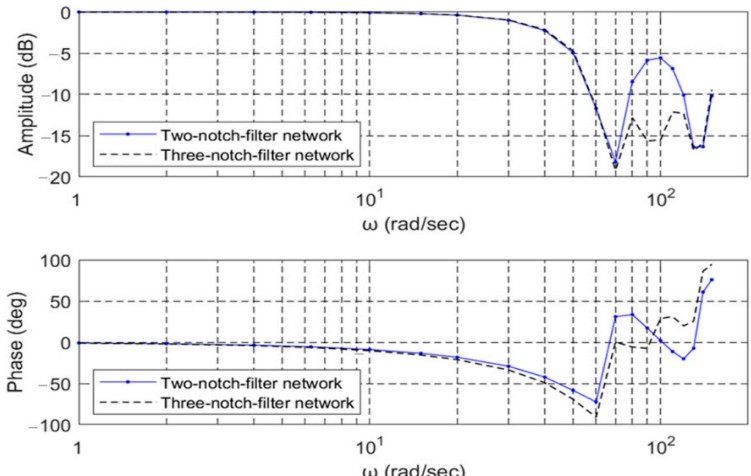

**Figure 17.** The bode plot of the notch filter networks.

The final compensated output $X_{L1}$ frequency response is shown in Figure 18, where the sixth-order simulation curve better fits the data than the fourth-order in most high frequency amplitude points. It is shown that the specification has been well satisfied. The response data at the marked frequencies is listed in Table 6.

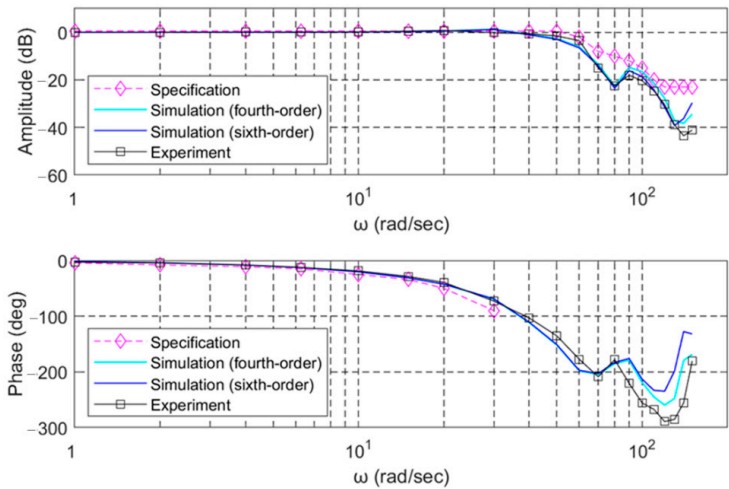

**Figure 18.** The final output $X_{L1}$ dynamics of the electro-hydraulic actuation system with the combined control algorithm.

**Table 6.** The compensated load dynamic response at the specified frequencies with the combined control algorithm.

| Frequency (Rad/s) | Amplitude (dB) | Phase (Degree) |
|---|---|---|
| 2 | 0.01 | −4.31 |
| 4 | 0.04 | −8.07 |
| 6.28 | 0.05 | −12.39 |
| 10 | 0.13 | −18.89 |
| 15 | 0.49 | −28.67 |
| 20 | 0.8 | −39.35 |
| 30 | −0.2 | −72.81 |
| 40 | −0.58 | −103.29 |
| 50 | −1.54 | −135.15 |
| 60 | −3.51 | −177.24 |
| 70 | −14.95 | −208.76 |
| 80 | −22.49 | −178.13 |
| 90 | −18.05 | −220.64 |
| 100 | −20.4 | −255.91 |
| 110 | −24.62 | −267.25 |
| 120 | −30.38 | −289.19 |
| 130 | −38.91 | −285.18 |
| 140 | −43.49 | −255.64 |
| 150 | −41.15 | −180.31 |

The relevant systems were tested in a series of successful launcher flights [25].

## 6. Discussion

As shown, unlike some applications where the load structural resonance can be ignored, the understanding and modeling of the controlled target dynamics are critical in highly dynamic actuation systems, especially for one to drive a complicated load with multiple DOFs. Due to the dominance of structural load dynamics as illustrated, hydro-mechanical resonance peaks emerge in the electro-hydraulic actuator piston loop, demonstrating their non-negligible effects and deserving modeling efforts, whatever an advanced controller might be used.

Since the demanding system bandwidth is approaching the structural natural frequencies and a frequency response is required as rigorous as specified by a set of amplitude and phase values at a series of given frequencies, a combined control algorithm has to be elaborately designed to accommodate specifications in the full cared frequency range. It was found that the separate use of a two-notch-filter network could not meet the stringent specification. Fortunately, with a digital control applied, the seemingly complicated algorithm can be easily implemented in software. Moreover, with the digital control, the pressure transducer or delicate devices can be eliminated in the critical control loop so that the reduced hardware cost and the higher inherent system reliability could be enjoyed. Additionally, with more studies, three or more notch filters might be applied.

One may argue that the system model might be simpler if the feedback is taken at the engine angular output point, because the structural load resonance items in the numerator outside the loop and denominator inside would be cancelled out and a full closed-loop is built. Nonetheless, there are the composite hydro-mechanical parts left, which are derived from the structural items. The modeling and control algorithm would be almost same.

In fact, the cascaded notch filter network can also be applied to actuators whose loads are described by ordinary one-mass-one-spring models, acting as a low-pass-high-resistance filter, which is usually welcome in engineering applications.

## 7. Conclusions

Multiple DOF models were presented to describe the thrust vector dynamics of a high thrust kerolox launch engine. A baseline fourth-order and a more accurate sixth-order transfer functions were developed out, with parameters identified by the actuator frequency

scanning tests and optimized by a full factor method. Normalized system models were given, comprising a closed-loop for the piston displacement including a fourth-order or sixth-order composite hydro-mechanical load effect, plus a fourth-order or sixth-order load structural resonance outside. In addition, the composite hydro-mechanical resonances were displayed in the open-loop position response. A combined control strategy comprising a notch filter network, a PID controller and a reduced-order feedforward compensation was used to obtain satisfactory dynamic performances in the whole interested frequency range, from low to high. The experiment data matched well with the simulations. It is demonstrated that, in a high-order dynamic aerospace electro-hydraulic actuation system where a high and stringent frequency response is required, both an appropriate high-order model and a delicate combined control strategy are enabling contributors to realize high performances.

**Author Contributions:** S.Z. put forward the methods to measure the load structural resonance and constructed the framework of the normalized model and the control algorithm; K.C. conducted the simulation and experiments and decided the specifics of a feedforward compensation; X.Z. built the original mathematical models and guided the simulation, experiments and substantiation of detailed control algorithms for a variety of actuation systems; Y.Z. optimized the model parameters of the structural dynamics and implemented the control algorithms for actuators; G.J. finalized the PID parameters; C.Y. developed the full factor optimization computer program for the control parameters and constructed the algorithm framework for digital signal processor based controllers; X.X. contributed to the data analysis and experiments. All authors have read and agreed to the published version of the manuscript.

**Funding:** This research received no external funding.

**Institutional Review Board Statement:** Not applicable.

**Informed Consent Statement:** Not applicable.

**Data Availability Statement:** The data presented in this study are available on request from the corresponding author.

**Acknowledgments:** The research has been funded by the Chinese non-toxic-non-pollution launch vehicle programs. This paper publication is supported in part by the specific program "Manufacturing Basic Technology and Critical Components" of the National Key Research and Development Program of China (No. 2019YFB2005101).

**Conflicts of Interest:** The authors declare no conflict of interest.

## Abbreviations

The following abbreviations are used in this manuscript:

| | |
|---|---|
| DPF | Dynamic Pressure Feedback |
| PID | Proportional, Integral and Differential |
| TVC | Thrust Vector Control |
| EHA | Electro-hydrostatic Actuator |
| DOF | Degree of freedom |
| D/A | Digital-to-Analog |
| A/D | Analog-to-Digital |

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
