# Peer review of "A High-Order Load Model and the Control Algorithm for an Aerospace Electro-Hydraulic Actuator†"

_actuators, doi:10.3390/act10030053_

Round 1

Reviewer 1 Report

In the manuscript entitled "A High-order Load Model and the Control Algorithm for an Aerospace Electro-Hydraulic Actuator" authors presents very interesting research efforts. In general the authors discuss the control model and control algorithm, also I must underline that the object (device is very interesting) and research task is ambitious. The article interested me very much, the mathematical model is prepared very carefully. Despite that he does not fully deal with actuators of this type, everything is understandable for me. I greatly appreciate and praise this diligence in preparing the "control model", the authors successive steps to build this model. I have only minor comments for discussion and inclusion in the modified version of this work.

My comments to discussion i put below.

  1. In general you use in the title of the paper "Electrohydraulic". In is correct use "Electro-hydraulic actuator" with dash. You must look to references written by English researchers.
  2. You also have quite a lot of articles from Asia in your literature. Perhaps it is worth adding a few references from world literature.
  3. When writing manuscript, I always wonder whether aberrations and list of symbols sections should be written at the beginning or at the end of the article.
  4. In my opinion, more useful in the beginning of manuscript. He does not suggest the order!
  5. In the row 110, the authors wrote "placement sensor imbedded inside the rod". In my opinion better is word "embedded".
  6. Is it possible to show a photo of the "engine" part in the article. This allows the reader to estimate the size of the device. Why the model needs to be so complex.
  7. I have comments to equations: (2), (3) and (4). Some symbols are written by normal font and some by italic font. This must be unified in manuscript.
  8. The same problem are in equation (8) and (9).
  9. I suppose that the idea of using the "two-mass-two-spring model" is very good. The device is so complex and there are complex phenomena that are simply not present in many control systems.
  10. I am interested in how the parameter "Lumped engine rotational inertia" in Table 1 was estimated. It was analytical calculated? It is possible experimentally measured? This parameter can have big impact on results.
  11. In the Figure 10, 11, 13 and 14, the "x" axis is marked by as a frequency and units are [rad/sec]. In general on the Bode Plots, the "x" axis is marked by "omega" and unit is [rad/sec]. Of course in manuscript body, it is allowed to use word "frequency", but on the Figures always we use "omega". In my opinion this must be modified before final publication.
  12. In my opinion the "search optimization method" is very easy to introduce. How look like the algorithm of such method. Do you use specific objective function? I ask, because in optimization procedure, we have design variables and constraints? The information about such parameters are not provided by authors. Please describe how your procedure works? In my opinion the research value will be bigger than you applied the "automatic procedure" managed by optimization procedure. You can read about such problems in manuscript entitled "Swarm-Based Design of Proportional Integral and Derivative Controllers Using a Compromise Cost Function: An Arduino Temperature Laboratory Case Study" published in Algorithms MDPI. You can also read manuscript "Self-Learning Salp Swarm Optimization Based PID Design of Doha RO Plant" from algorithm.
  13. In my opinion you should modified the reference list.You following phases "15. Mishra S K, Wrat G, Ranjan P, et al.", "G. Rath, E. Zaev and D. Babunski,". The word "and" and "et. al" is incorrect.
  14. Sometimes you use "Yin, C.; Zhao, S.; Chen, K." and sometimes "Yao J, Di D, Han J." The style second is incorrect in MDPI.

Reviewer 2 Report

Dear Authors,

Please find attached remarks to Your paper.

Reviewer 3 Report

This paper presents a controller for a rocket engine gimbal system.  The paper is clear enough and the English is good.  I am not an aerospace engineer so i can't comment on the significance of the application or whether the performance specifications are approprite, but the controller developed seems to meet these specifications.  Experimental results seem plausible and verify the analysis.

A few specific comments:

-better description of assumptions in the model development are required. E.g. the model ignore nonlinearities in friction and hydraulic valve (this is mentioned being significant later in lines 222-224, but should addressed in the model)

-should line 209 be "nearby" rather than "nearly"

-more x axis labels on log plots are required. E.g. 1,10,100 instead of just 1, in Fig 12.

-it appears that there will be a third resonant peak above the frequency range of Fig 12. Is this likely?  While this may be a small amplitude in displacement, it may be significant in velocity and acceleration

-Note that the plagiarism detector report shows large plagiarized sections, but these are just sections from the conference paper, so this should be ignored.

Reviewer 4 Report

In Figure 2, two coordinates X and XL are marked, and in equations (3) - (5) there are coordinates Xl1 and Xl2.

In the mathematical model described by equations (1) - (5), the servo valve and the Coulomb friction force in the mechanical elements were not taken into account.

The dynamic model does not take into account the equation of the cylinder motion.

The actuator is only used to load the 2-DoF dynamics system.

In this context, the caption under Figure 2 is inappropriate.

The coordinate differences concern not only the displacement (see elastic elements), but also their velocity (see damping elements).

Finally, a simplified linear dynamic model used in simple dynamic problems of automation systems was adopted.

From formula (4) we get (7) for XL1 = and not XL =.

Equations (6) and (7) should be presented as a transition function.

There is an error in the block diagram of Figure 4.

The manuscript presents the frequency characteristics of the electro-hydraulic actuation, although no dynamic equation of motion of a hydraulic actuator has been written.

Round 2

Reviewer 1 Report

The authors carefully prepared the revised manuscript. They take into account my all suggestion. I accept discussion. I have no further questions. I have considered all the questions and answers asked by the various reviewers. In my opinion, the authors put a lot of energy into improving the work. The level of discussion and preparation of modified manuscript was satisfactory. 

Reviewer 4 Report

I believe that the manuscript has been significantly improved and now warrants publication in Actuators.